# ROBUST PROBABILISTIC UNSUPERVISED SEGMENTATION WITH UNCERTAINTY MODELING

## ABSTRACT

Unsupervised semantic segmentation aims to assign a semantic label to each pixel in an image, identifying the object or scene class without any supervision. However, the task becomes particularly difficult due to factors like unclear or overlapping boundaries, intricate object textures, and the presence of multiple objects within the same region. Traditional unsupervised models often suffer from class misalignment and poor spatial coherence, leading to fragmented and imprecise segmentation, often employing postprocessing with Conditional Random Fields (CRFs) to improve their results. Additionally, deterministic models lack the ability to capture prediction uncertainty, making their outputs particularly prone to errors in ambiguous regions. To address these issues, we propose a probabilistic unsupervised semantic segmentation framework that enhances the robustness and accuracy of segmentation by refining predictions through uncertainty modeling and spatial smoothing techniques. We also introduce a novel loss function that encourages the model to focus on learning similarities within pixels by leveraging feature information from pre-trained vision transformer backbones. We also provide theoretical analyses of our proposed loss function, highlighting its favorable properties in relation to the optimization of our models. Our method demonstrates superior accuracy and calibration, outperforming various baselines across multiple unsupervised semantic segmentation benchmarks including COCO, Potsdam, and Cityscapes. In conclusion, our framework offers a foundation for more reliable, uncertainty-aware segmentation models, advancing research in unsupervised semantic segmentation.

## 1 INTRODUCTION

Semantic segmentation is a powerful tool in the field of computer vision, providing a complex and nuanced understanding of images down to the pixel level. As a dense prediction problem, semantic segmentation surpasses simple object detection and image classification by offering a detailed, granular perspective of an image's content. Unsupervised image segmentation builds upon this, by aiming to segment images without the use of labeled training data.

Despite recent progress in self-supervised and unsupervised segmentation techniques, considerable challenges remain. Noisy image regions or blurry edges, often caused by low resolution or motion blur, may result in regions of poor spatial coherence, leading models to output inconsistent and fragmented labels for adjacent parts of the same object. Additionally, images arising in real-world tasks often contain multiple instances of the same object class varying drastically in size and available contextual information. Models must be capable of capturing fine-grained details while still remaining capable of labeling large objects coherently. Moreover, variability in the appearance of the objects present in the images themselves, due to changes in color, lightning, or texture, renders it difficult to consistently identify and distinguish objects. Situations like this often arise in dense urban scenes, where objects may also overlap one another like people on sidewalks or trees in the background of an urban environment. Together, these challenges pose significant obstacles for robust and accurate segmentation models.

Existing unsupervised segmentation frameworks primarily leverage clustering algorithms, such as K-means (Lloyd, 1982; MacQueen et al., 1967), Fuzzy C-means (Bezdek et al., 1984), or Mean-Shift (Comaniciu & Meer, 2002), which group pixels based on similarity in color, texture, or intensity.

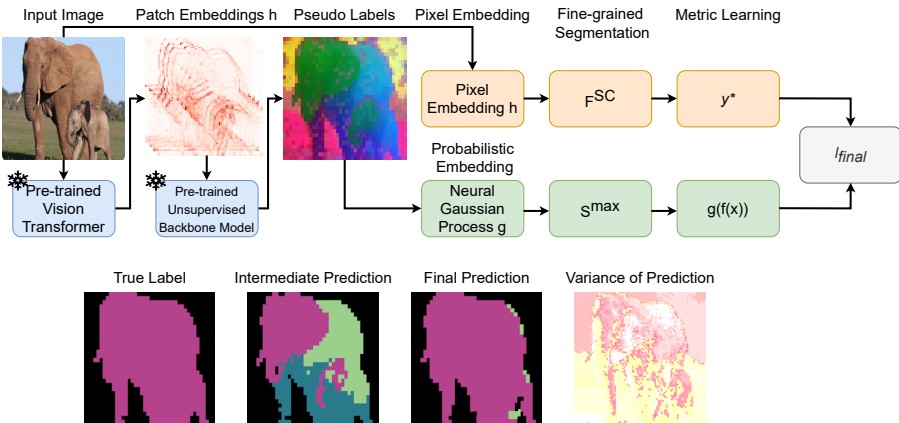

Figure 1: Illustration of our proposed probabilistic unsupervised segmentation framework. We can divide our modeling pipeline conceptually into five distinct steps: (i) Obtaining patch embeddings for each input image by passing them through a frozen ViT, (ii) Generating pseudo labels for the input image by passing the patch embeddings through a pretrained frozen Backbone Model (STEGO, EA-GLE, CAUSE, KNNs, etc.), (iii) Probabilistic refinement via modified SNGP for uncertainty-aware Gaussian embeddings, (iv) Fine grained segmentation to delineate objects, (v) Metric learning to categorize distinct object classes. Steps highlighted in blue signify frozen parts of the pipeline. Green and orange signify probabilistic and deterministic components respectively.
The four images on the bottom showcase from left to right: The true label (only used for evaluation), intermediate predictions produced during training, the final prediction after training has been concluded, the variance of the predictions, and dark red regions highlight regions of higher uncertainty.

However, these methods are inherently limited by the pixel-level locality of clustering approaches, which often struggle to capture broader contextual relationships within images.

In recent years, deep learning-based methods, such as autoencoders (Evan et al., 2020) and GANs (Goodfellow et al., 2014), have gained popularity for feature learning in segmentation tasks. Many newer techniques now combine deep learning with classical clustering and filtering methods. Prominent models like STEGO (Hamilton et al., 2023), EAGLE (Kim et al., 2024), HP (Seong et al., 2023), CutLER (Wang et al., 2023), SAM (Kirillov et al., 2023), U2Seg (Niu et al., 2024) and CAUSE (Kim et al., 2023) incorporate self-supervised, pretrained Vision Transformer (ViT) backbones like DINO (Caron et al., 2021) to extract features. These are then paired with smaller, trainable models to produce dense segmentation maps.

However, the segmentation maps generated by these models often suffer from noise, particularly in low-frequency regions, leading to grainy outputs. To mitigate this, postprocessing techniques such as clustering, upsampling, and refinement via fully connected Conditional Random Fields (CRFs) (Quattoni et al., 2004) are commonly applied.

Our approach differs fundamentally from these methods. Rather than applying additional postprocessing, we leverage the coarse segmentation maps produced by these models as pseudo-labels for our SNGP model. Moreover, existing deterministic models lack the capability to quantify uncertainty in their predictions, making them particularly vulnerable to errors in ambiguous or complex regions of the image.

To address these limitations, we propose a post-hoc probabilistic framework, combined with a novel loss function, which can be seamlessly integrated into existing methods (Kim et al., 2023; 2024; Hamilton et al., 2023). Our framework enhances semantic segmentation by making it uncertainty-aware, improving accuracy, and demonstrating robustness against noisy regions and blurry edges

Our contributions are: (i) We introduce a novel probabilistic framework followed by a novel loss function that enables a more precise definition of class boundaries and improves the robustness of prediction by unsupervised image semantic segmentation. (ii) Our distance-aware uncertainty loss and probabilistic embedding enable the model to better capture the underlying uncertainty in the data. By accounting for spatial relationships and variations in the embedding space, our method improves the

segmentation quality, particularly around ambiguous or overlapping class boundaries, while providing more reliable confidence estimates. (iii) We provide a theoretical foundation for the convergence of our proposed algorithm. Specifically, we show that our proposed loss function substantiates the convergence of the optimization process. The theoretical justification not only highlights the robustness of our approach but also provides a principled explanation for its effectiveness in improving representation quality. (iv) We conduct comprehensive experiments to show the advantage of our method compared to deterministic approaches. We test our method on three baselines and examine it on the Cityscapes, Potsdam, and COCO-Stuff segmentation datasets. Our approach achieves superior performance to uncertainty trade-off compared to the other methods, which highlights the promising benefit of probabilistic training to unsupervised segmentation frameworks.

## 2 BACKGROUND AND RELATED WORKS

**Learning Unsupervised Representation and Semantic Pseudo-label Assignment** Recent state-of-the-art unsupervised segmentation techniques exploit self-supervised learning models like DINO (Caron et al., 2018) as feature extraction for pixel embedding. These methods utilize pretrained Vision Transformer (ViT) models to generate rich, semantic representations that can be leveraged for downstream tasks. For instance, STEGO (Hamilton et al., 2023) introduces an approach to distill unsupervised features into semantic labels, achieving improved spatial coherence in segmentation results. EAGLE (Kim et al., 2024) enhances this by utilizing the eigenvectors of a similarity matrix-based Laplacian, allowing for the discovery of semantic relations between objects. Similarly, CAUSE (Kim et al., 2023) advances the field by incorporating a discretized concept clusterbook and concept-wise self-supervised learning to refine the segmentation process. These approaches focus primarily on label assignment by deriving pseudo-labels from high-level feature embeddings. Despite the progress, there remains a challenge in managing uncertainties and noise inherent in unsupervised methods. Our work addresses these limitations by proposing a probabilistic framework that integrates seamlessly with existing pseudo-labeling techniques, introducing a novel loss function to account for uncertainty in predictions. This not only improves the accuracy of segmentation but also makes it more robust to noisy and ambiguous boundaries.

**Spectral Normalized Gaussian Process** (SNGP) (Liu et al., 2020) improves the uncertainty quantification capabilities of deep learning models by introducing two key modifications: spectral normalization and a Gaussian process output layer. Spectral normalization regularizes the network's weights, ensuring smooth representations that preserve distances between input samples. The Gaussian process output layer models the uncertainty in predictions using a probabilistic framework. This makes SNGP suitable for **supervised learning tasks**, where the goal is to learn a mapping from input data to desired outputs. The SNGP model can be expressed as $f(x) = GP(\mu(x), k(x, x'))$, where $\mu(x)$ and $k(x, x')$ represent the mean and covariance functions of the GP, respectively. These functions are parameterized by a neural network, with spectral normalization applied to each layer to control the Lipschitz constant, ensuring $\|W\|_\sigma \leq 1$ where $W$ is the weight matrix of each layer and $\|.\|_\sigma$ denotes the spectral norm. Inspired by SNGP (Liu et al., 2020), we propose a Gaussian neural embedding to generate probabilistic embeddings at the pixel level to capture the uncertainty and variability in the data and expand SNGP for unsupervised setting.

## 3 METHOD

Our method integrates several key components to achieve robust segmentation result. First, it utilizes vision transformers for representation learning and feature extraction which result in patch embedding (3.1). The patch embedding are then passed through a pseudo-labeling process (using existing approaches such as EAGLE (Kim et al., 2024), STEGO (Hamilton et al., 2023), CAUSE (Kim et al., 2023), or KNNs) to assign labels to each pixel. To further refine these predictions, we introduce a probabilistic clustering refinement network (3.2). We leverage a modified SNGP method that generates Gaussian neural embeddings for each pixel and enables capturing the inherent uncertainty in the unsupervised setting. The pixel embeddings are then meticulously segmented through clustering, allowing for precise delineation and isolation of distinct elements within the image. Finally, metric learning facilitates the recognition and categorization of different object classes (3.4).

## 3.1 LEARNING REPRESENTATION

Given an unannotated dataset of training images, randomly sampled and defined as $\mathbf{X} = [\mathbf{x}_1, \ldots, \mathbf{x}_n]^N \in \mathbb{R}^{N \times D}$ and a transformation function $\tau$ that operates on this data. The transformation function plays a crucial role in improving the training process by generating augmented samples $\tilde{\mathbf{x}} \triangleq \tau(\mathbf{x})$ for each sample in $\mathbf{X}$. The augmentation process involves sampling $\tau$ from a distribution of suitable data transformations. Examples of such transformations include partially masking image patches Caron et al. (2021) or applying various image augmentation techniques Chen et al. (2020). The augmented views and original samples are then fed to an encoder network $f_{\boldsymbol{\theta}}$ with trainable parameters $\boldsymbol{\theta}$. The encoder (e.g., ResNet-50 He et al. (2016), ViT Dosovitskiy et al. (2021)) maps distorted samples to a set of corresponding features. We call the output of the encoder the *embedding*.

As depicted in Fig. 1, the extracted features are passed to the unsupervised pretrained segmentation model. Similar to Hamilton et al. (2023); Kim et al. (2024), We construct a lookup table for each image's $K$-Nearest Neighbors (K-NNs) $(x_{\text{knn}})$ based on cosine similarity within the feature space of the ViT backbone. This process produces pseudo-labeled images with semantic segmentation. The original images and pseudo-labeled images are then fed to a probabilistic framework.

## 3.2 PROBABILISTIC SEMANTIC SEGMENTATION

In the absence of ground-truth labels $y$, our probabilistic framework $g_{\Psi}$, parameterized by $\Psi$, operates on image pairs $(x^i)$ and their corresponding pseudo-labels $y_*^i$. To incorporate distance-aware uncertainty, we design the segmentation head using a Gaussian Process (GP) with a Laplace approximation. This approximation estimates the NGP posterior, providing the segmentation head with both the mean $\mu$ and variance $\sigma$. Similar to SNGP, we employ a fixed weight matrix $W_{L,D_L \times C}$ and a fixed bias vector $b_{L,D_L \times 1}$. For each logit, the GP prior is approximated via a neural network layer, using fixed hidden weights $W$ and learnable output weights $\beta_k$ (see Eq. 9).

## 3.3 FINE-GRAINED SEGMENTATION

As shown in Fig. 1, fine-grained segmentation is conducted through both deterministic and probabilistic pathways. By performing fine-grained clustering on either probabilistic or deterministic embeddings, we apply a K-Nearest Neighbors (KNN) approach enhanced with image randomization. This technique involves applying random augmentations to the input images during the KNN search, effectively increasing the diversity of the dataset and capturing a broader range of visual contexts. As a result, our method is capable of distinguishing even subtle object boundaries, leading to more precise and accurate segmentation, particularly in capturing fine-grained details.

## 3.4 METRIC LEARNING

To achieve end-to-end training, we formulate and train our algorithm by:

$$\mathcal{L}_{final} = \lambda^{self} \mathcal{L}_{total}\left(x, x, b^{self}, \lambda_{mc}, \lambda_{ce}, \lambda_{norm}\right) + \lambda^{knn} \mathcal{L}_{total}\left(x, x^{knn}, b^{knn}, \lambda_{mc}, \lambda_{ce}, \lambda_{norm}\right)$$
$$+ \lambda^{rand} \mathcal{L}_{total}\left(x, x^{rand}, b^{rand}, \lambda_{mc}, \lambda_{ce}, \lambda_{norm}\right)$$
$$(1)$$

where the $\mathcal{L}_{total}$ computed as:

$$\mathcal{L}_{total} = \lambda_{ce} \mathcal{L}_{ce}^*(x^m, x^n) + \lambda_{mc} \mathcal{L}_{mc}^*(x^m, x^n, b) + \lambda_{norm} \|\beta\|^2 \qquad (2)$$

in which $\mathcal{L}_{ce}$ and $\mathcal{L}_{mc}$ are pseudo-cross-entropy loss and maximum correlation loss respectively weighted by hyperparameters $\lambda_{ce}$ and $\lambda_{mc}$ to control their relative importance. We explain and provide more details in Section 4.2 and 4.2. The training and inference algorithms are presented in pseudo-code form in Figures 1 and 2, respectively. Importantly, we provide a computational complexity analysis in C.

# 4 THEORETICAL JUSTIFICATION

In this section, we present the mathematical foundation for our algorithm.

## 4.1 CONSTRUCTION OF NGP

Our NGP (Neural Gaussian Process) network serves as the semantic segmentation head, inspired by the Laplace approximation for neural Gaussian processes from Liu et al. (2020). Unlike standard dense output layers in self-supervised networks, our model incorporates a Gaussian Process (GP) with a Radial Basis Function (RBF) kernel. The posterior variance at a point $x$ is determined by the $L_2$ distance from the training data, using pseudo-labels generated by the backbone architecture. This allows the model to better capture uncertainty and produce more robust segmentation results. It also makes our model distance-aware uncertainty. We provide more detail in A. To efficiently estimate the GP, we employ the Random Fourier Feature (RFF) expansion, a computationally efficient method for approximating shift-invariant kernels Rahimi & Recht (2007).

The kernel is defined as:

$$k(\mathbf{x}, \mathbf{y}) = \int_{\mathbb{R}^d} \phi(\mathbf{x})^\top \phi(\mathbf{y}), d\mathbf{w}, \tag{3}$$

where $\mathbf{x}$ and $\mathbf{y}$ are data points, and $\phi(\mathbf{x})$ is a feature map projecting the input into a higher-dimensional space. RFF provides an efficient approximation by mapping the inputs to random features, rather than using the full kernel function.

For the Gaussian RBF kernel, the feature map is:

$$\phi(\mathbf{x}) = \sqrt{\frac{2}{m}} \cos(\mathbf{W}\mathbf{x} + \mathbf{b}), \tag{4}$$

where $\mathbf{W}$ is sampled from the Fourier transform of the kernel's distribution, and $\mathbf{b}$ is drawn uniformly from $[0, 2\pi]$. The approximation allows the inner product in the transformed space to estimate the original kernel efficiently.

Considering a dataset of training images $x^i$, randomly sampled with unknown $y^i$ and defined as $\mathcal{D}^* := \{x^i, y_*^i\}_{i=1}^N$. Let $y_*^i = \{y_{*,1}^i, \cdots, y_{*,D_L}^i\}$ with $y_{*,k}^i := 1_{\{k=\operatorname{argmax}_{i \in \{1, \cdots, D_L\}} h(f(x^i))\}}$ be the chosen class prediction of the backbone $f_\theta$ combined with its original final layer $h$ for the image $x^i$. The Gaussian process output layer $g_{N \times 1} = [g(f_{\theta_1}), \cdots, g(f_{\theta_i}), \cdots, g(f_{\theta_N})]^\top$ follows a multivariate normal distribution a priori:

$$g_{n \times 1} \sim \text{MVN}(\mathbf{0}_{N \times 1}, \mathbf{K}_{N \times N}), \tag{5}$$

where

$$\mathbf{K}_{i,j} = \exp(-\left\| f_{\theta_i} - f_{\theta_j} \right\|_2^2 / 2) \tag{6}$$

The posterior distribution is given by $p(g \mid \mathcal{D}^*) \propto p(\mathcal{D}^* \mid g) \, p(g)$, where $p(g)$ is the GP prior in (5) and $p(\mathcal{D}^* \mid g)$ is the data likelihood for classification (i.e., exponentiated pseudo-cross-entropy loss).

We follow the approximation method of (Liu et al., 2020), which applies the Laplace approximation to the RFF expansion of the GP posterior. Specifically, the GP prior is approximated using a low-rank factorization of the kernel matrix $\mathbf{K} = \Phi\Phi^\top$:

$$g_{N \times 1} \sim \text{MVN}\left(\mathbf{0}_{N \times 1}, \Phi\Phi_{N \times N}^\top\right), \Phi_{i, D_L \times 1} = \\ \sqrt{2/D_L} \, \cos\left(-\mathbf{W}_L f_i + \mathbf{b}_L\right), \tag{7}$$

where $\Phi$ is the feature map, $\mathbf{W}$ is a randomly initialized matrix, and $\mathbf{b}$ is a bias vector drawn uniformly from $[0, 2\pi]$. We apply spectral normalization to $\mathbf{W}$ for stabilization, as detailed in A.2.

The GP posterior is then approximated using:

$$\beta_k \mid \mathcal{D}^* \sim \text{MVN}\left(\hat{\beta}_k, \hat{\Sigma}_k\right), \hat{\Sigma}_k^{-1} = \mathbf{I} + \sum_{i=1}^N \hat{p}_{i,k}\left(1 - \hat{p}_{i,k}\right) \Phi_i \Phi_i^\top, \tag{8}$$

where $\hat{p}i, k = \text{softmax}(gk(f_i))$ represents the model prediction. During training, the posterior mean $\hat{\beta}$ is updated via stochastic gradient descent (SGD) over the log posterior, using the pseudo-cross-entropy loss. The precision matrix $\hat{\Sigma}_k$ is updated at the final epoch, and its inverse gives the posterior covariance.

## 4.2 Objective Function

**Pseudo Cross-Entropy**  For the $k^{\text{th}}$ logit, our approximation to the GP prior in (5) can be represented as a neural network layer with fixed hidden weights $\mathbf{W}$ and learnable output weights $\beta_k$. For each pixel $x_{h,w}^m$ of image $x^m$, this gives

$$g_k\left(f(x_{h,w}^m)\right) = \sqrt{2/D_L} \ \cos\left(-\mathbf{W}_L f(x_{h,w}^m) + \mathbf{b}_L\right)^\top \beta_k := \Phi_{mhw}^\top \beta_k, \tag{9}$$

where $\beta_k \sim N(0, \mathbf{I})$. The pseudo-cross-entropy loss is then defined as:

$$\begin{aligned}
\mathcal{L}_{ce}(\mathcal{D}^* \mid \beta) &:= -\log p(\beta \mid \mathcal{D}^*) \\
&= -\log p(\mathcal{D}^* \mid \beta) + \frac{1}{2}\|\beta\|^2 \\
&= -\sum_{k=1}^{D_L} \log p(\mathcal{D}^* \mid \beta_k) + \frac{1}{2}\|\beta\|^2,
\end{aligned} \tag{10}$$

where we can use the estimates of the Gaussian Process posterior mean ($\text{logit}(x) = \Phi^\top \beta$) and the backbone's prediction for each class $k \in \{1, \cdots, K\}$:

$$\mathcal{L}_{ce}^*(\mathcal{D}^* \mid \beta_k) := -\log p(\mathcal{D}^* \mid \beta_k) = -\sum_{m=1}^{N} \sum_{hw} 1_{\{k=\arg\max_{i \in \{1, \cdots, D_L\}} h(f(x_{h,w}^m))\}} \ \log\left(\Phi_{mhw}^\top \beta_k\right) \tag{11}$$

By incorporating this loss function and using the Gaussian Process output layer, our model is guaranteed the same properties as the supervised version of SNGP regarding the pseudo-labels instead of the true labels. This means that the predictions of the backbone combined with the GP output layer will converge towards the distribution of the backbone with its original final layer.

**Maximum Correlation Loss**  To improve semantic segmentation, we introduce a Maximum Correlation (MC) loss, which leverages self-feature correspondence tensors. For two feature tensors $f$ and $g$ in $\mathbb{R}^{CHW}$, we define the self-feature correspondence tensor as:

$$F_{hwij} := \sum_c \frac{f_{chw}}{|f_{hw}|} \frac{g_{cij}}{|g_{ij}|} \tag{12}$$

The tensor measures the correlation between spatial positions $(h, w)$ and $(i, j)$, capturing relationships between the channels that aid in segmentation. To incorporate the GP predictions, we use the softmax of the logits and compute the index of the maximum predicted class for each pixel:

$$k_{max}^{mij} = \arg\max_k \sigma_k(\Phi_{mij}^\top \beta) \tag{13}$$

The MC loss function rewards similar pixel pairs and penalizes dissimilar pairs, and is defined as:

$$\mathcal{L}_{mc}(x^m, x^n) = -\frac{1}{HW} \sum_{hwij} F_{hwij} S_{max}(x_{i,j}^m, x_{h,w}^n) \tag{14}$$

where $S_{max}$ represents the maximum predicted class similarity between pixels. For practical optimization, we shift this loss to a non-negative range by adding a constant. Spatial centering is applied for balanced training Hamilton et al. (2023).

**Total Loss**  The final loss function, $\mathcal{L}_{total}$ (Eq. 2), combines the pseudo-cross-entropy and MC losses, with tunable weights $\lambda_{ce}$, $\lambda_{mc}$, and $\lambda_{norm}$ to optimize performance.

## 5 Experiments and Results

In this section, we first outline the implementation details, including dataset configurations, evaluation protocols, and specific experimental settings. Next, we assess our proposed method both qualitatively and quantitatively, ensuring a fair comparison with existing state-of-the-art approaches.

Additionally, we validate the effectiveness of our method through an ablation study. For further information, please refer to the supplementary material.

**Implementation Details** We utilize the DINO (Caron et al., 2021) pretrained vision transformer, which remains frozen throughout the training process for feature extraction. The training datasets are resized and five-cropped to a dimension of $244 \times 244$ pixels. For the probabilistic segmentation head, we employ two layers of Neural Gaussian Process (NGP) with ReLU activation. Our method can be easily integrated into other frameworks; in this study, we generate pseudo-labels using STEGO, EAGLE, and CAUSE. All backbones have an embedding dimension of 512. The source code is available at `https://anonymous.4open.science/r/PUIS2024-0010/README.md`

**Datasets** We evaluate our model on three datasets for unsupervised semantic segmentation: (1) COCO-Stuff(Caesar et al., 2018), which features detailed pixel-level annotations that enable a comprehensive understanding of various objects; (2) Cityscapes(Cordts et al., 2016), which captures diverse urban street scenes; and (3) Potsdam-3 (Ji et al., 2019), consisting of satellite imagery. Following the class selection protocols established in earlier studies (Hamilton et al., 2023; Cho et al., 2021; Ji et al., 2019; Kim et al., 2024; 2023), we utilize 27 classes from both the COCO-Stuff and Cityscapes datasets, while for the Potsdam-3 dataset, we include all three classes.

**Evaluation Metrics for Quantitative Results** We report the performance of our method using the following metrics: Top-1 Accuracy ↑: This metric refers to the proportion of pixels in the test set that the model correctly predicts as belonging to the correct class. In the context of semantic segmentation, it measures how many pixels are assigned the correct label, providing an overall indication of segmentation quality. Mean Intersection over Union (mIoU) ↑: This metric quantifies the overlap between the predicted segmentation and the ground truth. It is calculated as the average of the intersection over union for all classes, defined as the ratio of the intersection area to the union area. This provides a robust measure of the model's accuracy in delineating different object regions.

**Compared Methods** In our evaluation, we benchmark our method against three distinct baseline pseudo-labeling approaches: STEGO (Hamilton et al., 2023), EAGLE (Kim et al., 2024), and CAUSE (Kim et al., 2023). We integrated our method by replacing the projection head of each baseline with our probabilistic framework, allowing us to assess uncertainty in prediction. Additionally, we compared our approach to existing benchmark models across the various datasets. The performance results presented were averaged over five independent runs to ensure robustness and reliability.

**Results and Discussion** Based on the results obtained in Table 3, our method achieves superior performance to the state-of-the-art STEGO Hamilton et al. (2023) on the 3 classes of the Potsdam dataset, outperforming the next best baseline by 5% unsupervised accuracy. Our method achieved state-of-the-art performance on the Potsdam dataset. Based on the qualitative results shown in Fig. 2a, our algorithm distinguishes between different classes such as buildings and vegetation. By comparing our results to the backbone, our model appears to offer an improved delineation of the various classes, with a reduction in misclassified areas. The boundaries between buildings and vegetation are more accurately rendered, and there is a visible decrease in the fragmentation of predicted areas, suggesting a more consistent understanding of the spatial context. Based on the qualitative results obtained in Fig. 2b, our method produces segmentations that capture small objects and fine details.

On the Cityscapes dataset, our method successfully identifies people, streets, sidewalks, cars, and street signs with high detail and fidelity.Specifically, our method achieved an accuracy of 91.0% and a mean Intersection over Union (mIoU) of 28.1%, surpassing recent state-of-the-art methods such as STEGO model, which had an accuracy of 73.2% and a mIoU of 21.0%, and CAUSE algorithm which scored 89.8% in accuracy and 29.9% in mIoU.

The results obtained in Tables 3, demonstrate the advantages of using Vision Transformers (ViT) for unsupervised segmentation. Traditional unsupervised segmentation techniques, such as Random CNNs and SIFT, show relatively low performance, with accuracies around 38.2%. While more recent CNN-based models, like IIC (65.1%) and (Doersch et al., 2016) (49.6%), improve accuracy but still fall behind the ViT-based approaches. When DINO is combined with models like STEGO (77.0%), EAGLE (83.3%), and HP (82.4%), the results see a significant boost. More importantly, our probabilistic approach achieves the highest accuracy of 83.9%, highlighting its ability to refine segmentation results by leveraging probabilistic embeddings, even outperforming other state-of-the-art methods.

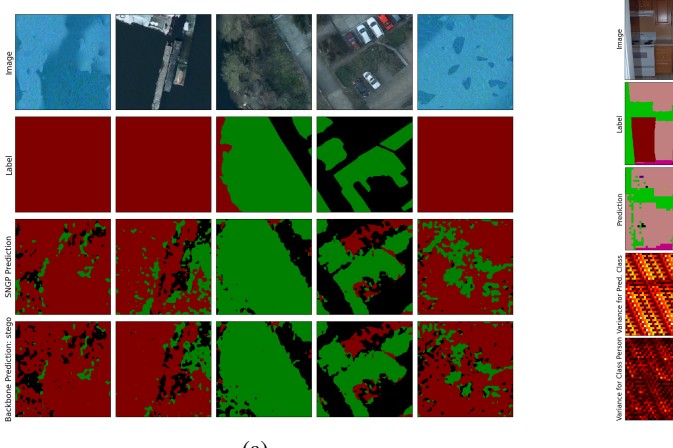 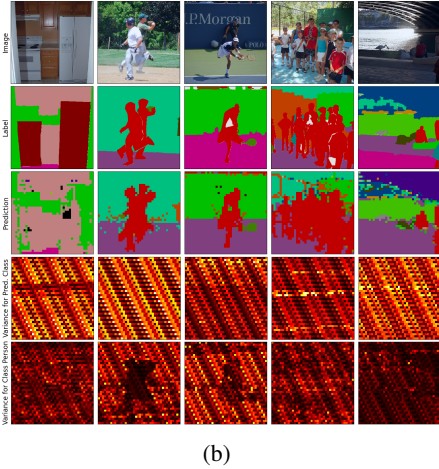

(a)                                                      (b)

Figure 2: (a) Comparison of ground truth labels and our predictions, and backbone for images from the Potsdam dataset, (b) Qualitative assessment of segmentation model performance on COCO-Stuff dataset. Top to bottom: original images, ground truth, our predictions, and variance for predicted pixel class and class expression, illustrating the model's uncertainty in its predictions

Table 1: Comparison of Accuracy and mIoU on the **COCO-Stuff** Validation data set.

| Model | Backbone | Unsupervised | | Linear Probe | |
|---|---|---|---|---|---|
| | | **Accuracy** | **mIoU** | **Accuracy** | **mIoU** |
| ResNet50 He et al. (2016) | ResNet50 | 24.6 | 8.9 | 41.3 | 10.2 |
| MoCoV2 Chen et al. (2020) | ResNet50 | 25.2 | 10.4 | 44.4 | 13.2 |
| Deep Cluster Caron et al. (2018) | ResNet50 | 19.9 | - | - | - |
| Doersch et al. Doersch et al. (2016) | ResNet18 | 23.1 | - | - | - |
| IIC Ji et al. (2019) | ResNet18 | 21.8 | 6.7 | 44.5 | 8.4 |
| MDC Cho et al. (2021) | ResNet18 | 32.2 | 9.8 | 48.6 | 13.3 |
| PiCIE Cho et al. (2021) | ResNet18 | 48.1 | 13.8 | 54.2 | 13.9 |
| DINO Caron et al. (2021) | ViT-S/8 | 30.5 | 9.6 | 66.8 | 29.4 |
| + STEGO Hamilton et al. (2023) | ViT-S/8 | 56.9 | 28.2 | 73.6 | 33.3 |
| + HP Seong et al. (2023) | ViT-S/8 | 57.2 | 24.6 | 75.6 | 42.7 |
| + CAUSE Kim et al. (2023) | ViT-B/8 | 69.6 | 32.4 | 78.8 | 47.2 |
| + EAGLE Kim et al. (2024) | ViT-B/8 | 64.2 | 27.2 | 76.8 | 43.9 |
| + Ours | ViT-B/8 | **71.1** | **33.0** | **80.5** | **52.3** |

The variance maps are especially insightful, as they provide a visual representation of the model's confidence in its predictions. High variance areas, especially at the edges of objects, could guide further model refinement, suggesting that additional features or training on edge-detection could improve performance.

The results from Table 1,3,2, strongly demonstrate the effectiveness of our probabilistic embedding framework in achieving state-of-the-art performance. Across multiple segmentation tasks, our method consistently outperforms existing approaches.

**Ablation Study**    To explore the effectiveness of our proposed method, we conducted a series of ablation studies dissecting various aspects of its design. Specifically, we examined: i) how changes to the loss function hyperparameter $\lambda$ influence performance. ii) the effect of varying the embedding space dimension $D_L$ on the model's behavior. iii) impact of different pseudo-labeling techniques. Furthermore, we provide an additional ablation analysis focusing on computational efficiency (see C) and more qualitative results.

Based on Fig. 3-a, by increasing the dimensionality of the probabilistic embedding the performance improved significantly. This suggests that higher-dimensional embeddings can capture more complex patterns in the data.

Fig. 3-b shows our ablation analysis on the hyperparameters of the loss function $\lambda_{mc}$ (maximum correlation) and $\lambda_{ce}$ (cross-entropy) loss significantly influences the model's performance. A combined loss approach appears to be beneficial, with the best results seen when the cross-entropy loss is given

Table 2: Comparison of Accuracy and mIoU on **Cityscapes** Validation data set.

| Model | Backbone | Unsupervised | |
|---|---|---|---|
| | | Accuracy | mIoU |
| | | Accuracy | mIoU |
| IIC Ji et al. (2019) | ResNet18 | 47.9 | 6.4 |
| MDC Cho et al. (2021) | ResNet18 | 40.7 | 7.1 |
| PiCIE Cho et al. (2021) | ResNet18 | 65.5 | 12.3 |
| DINO (Caron et al., 2021) | ViT-S/8 | 34.5 | 10.9 |
| + HP (Seong et al., 2023) | ViT-S/8 | 80.1 | 18.4 |
| + EAGLE (Kim et al., 2024) | ViT-S/8 | 81.8 | 19.7 |
| + Ours | ViT-S/8 | 82.1 | 19.9 |
| STEGO Hamilton et al. (2023) | ViT-B/8 | 73.2 | 21.0 |
| EAGLE Kim et al. (2024) | ViT-B/8 | 79.4 | 22.1 |
| + Ours | ViT-B/8 | 81.8 | 21.2 |
| CAUSE Kim et al. (2023) | ViT-B/14 | 89.8 | **29.9** |
| + Ours | ViT/B14 | **91.0** | 28.1 |

Table 3: Comparison of Top-1 Accuracy on the **Potsdam-3** data set.

| Model | Backbone | Unsup. Acc. |
|---|---|---|
| Random CNN (Ji et al., 2019) | ResNet18 | 38.2 |
| K-Means (Pedregosa et al., 2018) | ResNet18 | 45.7 |
| SIFT Lowe (1999) | ResNet18 | 38.2 |
| (Doersch et al., 2016) | ResNet18 | 49.6 |
| (Isola et al., 2015) | ResNet18 | 63.9 |
| Deep Cluster (Caron et al., 2018) | ResNet50 | 41.7 |
| IIC (Ji et al., 2019) | ResNet18 | 65.1 |
| DINO (Caron et al., 2021) | ViT-B/8 | 53.0 |
| + STEGO (Hamilton et al., 2023) | ViT-B/8 | 77.0 |
| + EAGLE (Kim et al., 2024) | ViT-B/8 | 83.3 |
| + HP (Seong et al., 2023) | ViT-B/8 | 82.4 |
| + Ours | ViT-B/8 | **83.9** |

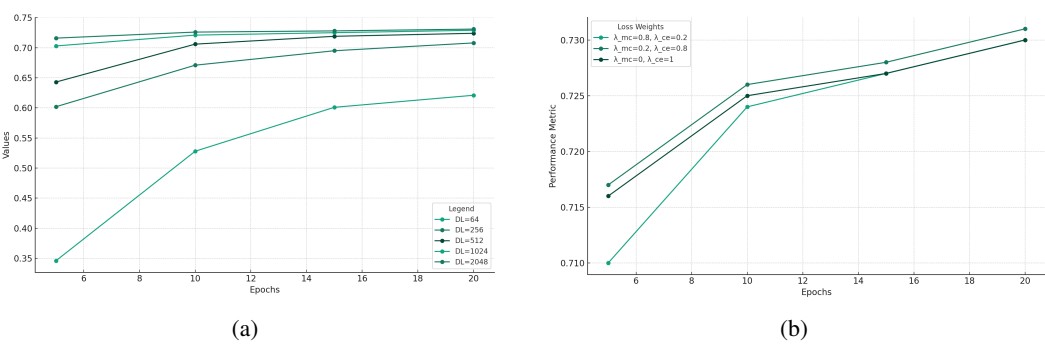

(a)          (b)

Figure 3: Study of hyperparameters of our proposed method (a) dimension of probabilistic embedding $(D_L)$, (b) $\lambda$.

more weight. This information can guide the fine-tuning of loss function weights for optimal model training outcomes.

Based on quantitative results obtained in Table 4, our probabilistic framework consistently boosts accuracy across all three models, with the most significant improvement observed in STEGO. The mIoU improvements are smaller but consistent, indicating that our method not only enhances overall pixel classification but also slightly refines the prediction of class boundaries. These results underscore the utility of integrating uncertainty-aware approaches in unsupervised segmentation tasks.

Table 4: Comparison of achieved Accuracy and mIoU on the **Cityscapes** with different pseudo-labeling techniques while all models utilized ViT-B/8 as the backbone.

| Model | Unsupervised | |
|---|---|---|
| | **Accuracy** | **mIoU** |
| STEGO (Hamilton et al., 2023) | 73.2 | 21.0 |
| + Ours | **81.6** | **21.8** |
| EAGLE (Kim et al., 2024) | 79.4 | 22.1 |
| + Ours | **82.0** | **22.2** |
| CAUSE(Kim et al., 2023) | 90.8 | 28.0 |
| + Ours | **91.0** | **28.1** |

## 6 CONCLUSION

In this paper, we presented a novel approach to probabilistic unsupervised segmentation that uses Gaussian embeddings and a newly designed loss function to provide a more robust representation for the task of unsupervised semantic segmentation. We also presented a theoretical analysis to justify the convergence properties of the proposed loss function. In addition, we demonstrated that our post-hoc framework and loss function can be integrated with existing deterministic unsupervised segmentation methods to improve both performance and robustness. Our experimental results confirm that our approach achieves state-of-the-art performance, demonstrating its potential for improving accuracy and stability in unsupervised segmentation tasks.

**Limitation and Future Work**    A limitation of our work lies in the dimensionality of the probabilistic embeddings. While our results indicate that increasing the dimensionality improves performance, likely due to the model's ability to capture more complex patterns and subtleties in the data, this comes with potential trade-offs. Higher-dimensional embeddings increase computational costs and may also lead to overfitting, particularly when embedding sizes become very large. Future work should investigate these trade-offs to determine optimal embedding sizes that balance model complexity with computational efficiency and generalization ability.

In future work, it would be interesting to develop our probabilistic framework for other tasks such as object detection. Extending this approach could allow for more robust uncertainty quantification in object localization and classification. Additionally, adapting the framework for tasks like video analysis might provide deeper insights into complex temporal and spatial dependencies. This extension could also help address challenges in real-time applications where uncertainty estimation is crucial, such as autonomous driving or medical imaging, where the framework's probabilistic nature can lead to more reliable decision-making.

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

# A    ADDITIONAL THEORY

In this part of the appendix, we focus on diving deeper into the theoretical background of our proposed method.

## A.1    DISTANCE AWARENESS IN DEEP LEARNING

In Liu et al. (2020) the authors provide a definition for Input Distance Awareness:

**Definition A.1.1** *(Input Distance Awareness) Let $p(y \mid \mathbf{x})$ be a predictive distribution trained on a domain $\mathcal{X}_{IND} \subset \mathcal{X}$, where $(\mathcal{X}, \|\cdot\|_X)$ is the input data manifold equipped with a suitable metric $\|.\|_X$. $p(y \mid \mathbf{x})$ is input distance aware if there exists $u(\mathbf{x})$ a summary statistic of $p(y \mid \mathbf{x})$ that quantifies model uncertainty (e.g., entropy or predictive variance) that quantifies the distance between $\mathbf{x}$ and the training data with respect to $\|.\|_X$, i.e.,*

$$u(\mathbf{x}) = v\left(d\left(\mathbf{x}, \mathcal{X}_{IND}\right)\right) \tag{15}$$

*where $v$ is a monotonic function and $d\left(\mathbf{x}, \mathcal{X}_{IND}\right) = E_{\mathbf{x}' \sim \mathcal{X}_{IND}} \|\mathbf{x} - \mathbf{x}'\|_X^2$ is the distance between $\mathbf{x}$ and the training data domain.*

It is worth noting that a Gaussian process (GP) equipped with a radial basis function (RBF) kernel satisfies this property. The predictive distribution of such a GP, denoted as $p(y \mid x) = \mathrm{softmax}(g(x))$, represents a softmax transformation of the GP posterior $g \sim GP$ under the cross-entropy likelihood. Additionally, the predictive uncertainty of the GP can be expressed using the posterior variance $u(x^*) = \mathrm{var}(g(x^*)) = 1 - k^\top V k^*$, where $k_i^* = \exp\left(-(1/2l) |x^* - x_i|^2 \ X\right)$, and $V_{N \times N}$ is a fixed matrix determined by the data. In contrast, this property is not guaranteed to hold for classical deep learning models. For a more detailed explanation, we recommend the second chapter of Liu et al. (2020).

## A.2    SPECTRAL NORMALIZATION

When the output layer $g$ is replaced with a Gaussian process, it allows the model $\mathrm{logit}(\mathbf{x}) = g \circ h(\mathbf{x})$ to become aware of the distance in the hidden space $|h(\mathbf{x}_1) - h(\mathbf{x}_2)|_H$. To ensure the meaningful correspondence between distances in the hidden space $|h(\mathbf{x}) - h(\mathbf{x}')|_H$ and the input space $|\mathbf{x} - \mathbf{x}'|_X$, it is crucial that the hidden mapping $h$ is distance preserving. Many modern deep learning models, such as ResNets and Transformers, are composed of residual blocks. Specifically, the hidden mapping can be expressed as $h(\mathbf{x}) = h_{L-1} \circ \cdots \circ h_2 \circ h_1(\mathbf{x})$, where $h_l(\mathbf{x}) = \mathbf{x} + g_l(\mathbf{x})$.

To ensure distance preservation, a simple method is employed by bounding the Lipschitz constants of all nonlinear residual mappings $\{g_l\}_{l=1}^{L-1}$ to be less than 1. This constraint guarantees that the distances in the hidden space remain consistent with the distances in the input space, thus preserving meaningful relationships between the data points. This result can be formally seen below:

**Proposition 1** *(Lipschitz-bounded residual block is distance preserving) Let $h : \mathcal{X} \to \mathcal{H}$ be a hidden mapping with residual architecture $h = h_{L-1} \circ \ldots h_2 \circ h_1$ where $h_l(\mathbf{x}) = \mathbf{x} + g_l(\mathbf{x})$. If for $0 < \alpha \leq 1$, all gl's are $\alpha$-Lipschitz, i.e., $\|g_l(\mathbf{x}) - g_l(\mathbf{x}')\|_H \leq \alpha \|\mathbf{x} - \mathbf{x}'\|_X \quad \forall (\mathbf{x}, \mathbf{x}') \in \mathcal{X}$. Then:*

$$L_1 \|\mathbf{x} - \mathbf{x}'\|_X \leq \|h(\mathbf{x}) - h(\mathbf{x}')\|_H \leq L_2 \|\mathbf{x} - \mathbf{x}'\|_X, \tag{16}$$

*where $L_1 = (1 - \alpha)^{L-1}$ and $L_2 = (1 + \alpha)^{L-1}$, i.e., $h$ is distance preserving.*

Proof is provided in Appendix E.1 of Liu et al. (2020). To maintain the distance-preserving property of the hidden mapping $h$, it is sufficient for the weight matrices of the nonlinear residual block $g_l(\mathbf{x}) = \sigma\left(\mathbf{W}_l \mathbf{x} + \mathbf{b}_l\right)$ to have a spectral norm (i.e., the largest singular value) less than 1, since $\|g_l\|_{\mathrm{Lip}} \leq \|\mathbf{W}_l \mathbf{x} + \mathbf{b}_l\|_{\mathrm{Lip}} \leq \|\mathbf{W}_l\|_2 \leq 1$.

To satisfy this Lipschitz constraint on $g_l$, spectral normalization (SN) is applied to the weight matrices $\{\mathbf{W}_l\}_{l=1}^{L-1}$, following the methodology presented in Liu et al. (2020). During each training iteration,

first the spectral norm $\hat{\lambda} \approx \|\mathbf{W}_l\|_2$ is approximated using the power iteration method, as outlined in Gouk et al. (2020) and Miyato et al. (2018). Subsequently, the weights are normalized according to:

$$\mathbf{W}_l = \begin{cases} c\mathbf{W}_l/\hat{\lambda} & \text{if } c < \hat{\lambda} \\ \mathbf{W}_l & \text{otherwise.} \end{cases} \tag{17}$$

Here, $c > 0$ acts as a hyperparameter to fine-tune the exact spectral norm upper bound on $\|\mathbf{W}_l\|_2$ such that $\|\mathbf{W}_l\|_2 \leq c$. This offers a level of adaptability, which is especially valuable when other regularization techniques like Dropout and Batch Normalization are in play, which could alter the Lipschitz constant of the original residual mapping as discussed in Gouk et al. (2020). Equation (17) thus grants us greater flexibility in managing the spectral norm, allowing it to better harmonize with the architecture in use.

## B  TRAINING AND PREDICTION ALGORITHMS

The pseudo-code for our proposed training and prediction methods is presented in Sections  1 and  2, respectively.

---

**Algorithm 1** Training

---

**Require:** Minibatches $\mathcal{D} = \{x^i\}_{i=1}^N$
**Ensure:** Estimated parameters $\beta_k$
  1: Initialization: $\hat{\Sigma} = \mathbb{I}$, $\mathbf{W}_L \overset{\text{iid}}{\sim} N(0,1)$ and $\mathbf{b}_L \overset{\text{iid}}{\sim} U(0, 2\pi)$
  2: **for** epoch $= 1$ **to** num_epochs **do**
  3:     **for** $i = 1$ **to** $N$ **do**
  4:         Get $x^{knn}$ and $x^{rand}$ for $x^i$
  5:         **for** $j = 1$ **to** num_pixel **do**
  6:             Compute feature correspondence tensor $F_{ij}^{SC}$ and pseudo-labels $y_{ij}^*$ with the backbone
  7:             Compute the output logits for each $k \in \{1, \cdots, D_L\}$ using Equation 9
  8:         **end for**
  9:         Compute the loss using Equation 2
 10:         Update the parameters $\beta, \{\mathbf{W_1}\}_{l=1}^{L-1}, \{\mathbf{b_1}\}_{l=1}^{L-1}$ using stochastic gradient descent
 11:         Perform Spectral Normalization (described in A.2) of $\{\mathbf{W_1}\}_{l=1}^{L-1}$
 12:         **if** final_epoch **then**
 13:             Update precision matrix $\{\hat{\Sigma}_k^{-1}\}_{k=1}^K$
 14:         **end if**
 15:     **end for**
 16: **end for**
 17: Compute posterior covariance $\hat{\Sigma}_k = \text{inv}\left(\hat{\Sigma}_k^{-1}\right)$

---

---

**Algorithm 2** Prediction

---

**Require:** Testing example pixel $\mathbf{x}$
  1: Compute Feature: $\Phi = \Phi_{D_L \times 1} = \sqrt{\frac{2}{D_L}} \cos\left(-\mathbf{W}_L f(\mathbf{x}) + \mathbf{b}_L\right)$
  2: Compute Posterior Mean: $\text{logit}_k(\mathbf{x}) = \Phi^\top \beta_k$
  3: Compute Posterior Variance: $\text{var}_k(\mathbf{x}) = \Phi^\top \hat{\Sigma}_k \Phi$
  4: Compute Predictive Distribution: $p(y \mid \mathbf{x}) = \int_{m \sim N(\text{logit}(\mathbf{x}), \text{var}(\mathbf{x}))} \text{softmax}(m)$

---

## C  COMPUTATIONAL COMPLEXITY

In this section, we turn our attention to assessing the computational complexity of both the training (displayed in Algorithm 1) and the prediction (displayed in Algorithm 2) algorithms as well as the loss function itself. Therefore, let $N \in \mathbb{N}$ be the number of samples within a minibatch and let $H, W \in \mathbb{N}$ be the height and width of an image. Let $x_{h,w}^m$ be the pixel at location $(h, w) \in \mathbb{N}^2$ of image $m \in \{1, \cdots, N\}$.

## C.1   Loss Function

To gain a comprehensive understanding of our model's computational requirements, we first dissect the components of the loss function, starting with the Maximum Correlation segment of the combined loss function, denoted by Equation **??**.

For the sake of this discussion, we will assume that the complexity for obtaining predictions from the backbone architecture is $\mathcal{O}(1)$. An examination of the backbone predictions' complexity will be presented in the subsequent section.

Let $g_k(f(x_{h,w}^m)) = \sqrt{2/D_L} \cos\left(-\mathbf{W}_L f(x_{h,w}^m) + \mathbf{b}_L\right)^\top \beta_k$ be the prediction from the SNGP head for the pixel $x_{h,w}^m$, as defined in Equation (9). We derive its computational complexity as follows:

$$\mathcal{O}(-\mathbf{W}_L f(x_{h,w}^m)) = \mathcal{O}(f^*(x_{h,w}^m)) = \mathcal{O}(D_L \times C) \tag{18}$$

$$\sqrt{2/D_L} \cos\left(f^*(x_{h,w}^m) + \mathbf{b}_L\right)^\top \beta_k = \mathcal{O}(D_L) \tag{19}$$

In the first computational step, we perform a matrix-vector multiplication between $\mathbf{W}_L$ of size $D_L \times C$ and $f(x_{h,w}^m)$ of size $C \times 1$. This operation has a computational complexity of $\mathcal{O}(D_L \times C)$. The following step incorporates a set of cosine and addition operations, each having a complexity of $D_L$, as well as a matrix-vector multiplication with $\beta_k$ of dimensions $D_L \times 1$. The latter operation has a complexity of $\mathcal{O}(D_L \times C)$. In total, we find that the overall complexity for generating class predictions $k \in \{1, \cdots, D_L\}$ is given by:

$$\mathcal{O}(g_k\left(f(x_{h,w}^m)\right)) = \mathcal{O}(D_L(C + 1 + 1)) = \mathcal{O}(D_L \times C) \tag{20}$$

The next relevant calculation is to determine the complexity of $\mathcal{O}(S_k(x_{i,j}^m))$ and $\mathcal{O}(k_{max}^{mij})$ for a single image pixel. In subsequent computations, each pixel pair undergoes six operations: computing $k_{max}^{mij}$ twice and retrieving their corresponding predictions for each image once. Given that these operations only involve linear-time computations $\mathcal{O}(1)$, such as the argmax and the softmax functions, the complexity for these operations is:

$$\mathcal{O}(S_k(x_{i,j}^m)) = \mathcal{O}(k_{max}^{mij}) = \mathcal{O}(D_L) \tag{21}$$

Since $k_{max}^{mij}$ and its corresponding $S_k(x_{i,j}^m)$ have already been computed, the calculation of $S_{max}(x_{i,j}^m, x_{h,w}^n)$ amounts to a computational complexity of $\mathcal{O}(1)$. Taking an image pair into account, the cumulative complexity for this stage can be formulated as:

$$\mathcal{O}((D_L \times C + 6D_L) \times H \times W) = \mathcal{O}(D_L \times C \times H \times W) \tag{22}$$

Next, we describe the complexity analysis for $F_{hwij}$ as outlined in Equation (12). This computation entails a multiplication across all pixels of an image pair, as well as a summation over the penultimate layer $C$ of the backbone architecture. Therefore, the resulting complexity can be expressed as:

$$\mathcal{O}(F_{hwij}) = \mathcal{O}(C \times H \times W \times H \times W) = \mathcal{O}(C \times (HW)^2) \tag{23}$$

The loss function $\mathcal{L}_{max}(x^m, x^n)$ for a pair of images, as described in Equation (**??**), comprises solely linear operations with a computational complexity of $\mathcal{O}(1)$. Given that all the necessary inputs have already been computed, we can formulate the computational complexity for this loss function as:

$$\mathcal{O}(\mathcal{L}_{mc}(x^m, x^n)) = \mathcal{O}((HW)^2) \tag{24}$$

Consequently, we get the overall computational complexity for the Maximum Correlation segment of the combined loss function, denoted by Equation **??**:

$$\mathcal{O}(D_L \times C \times H \times W + C \times (HW)^2 + (HW)^2) \tag{25}$$

For our second loss component, we examine the pseudo-cross-entropy loss as detailed in Equation 10.

Here, we also need to calculate the predictions $g_k(f(x^m_{h,w}))$ derived from Equation (9). Given those predictions for all classes $k \in \{1, \cdots, D_L\}$ need to be computed, the computational complexity for a single pixel in an image is:

$$\mathcal{O}(\sum_{k=1}^{D_L} g_k\left(f(x^m_{h,w})\right)) = \mathcal{O}(D_L(C + 1 + 1)) = \mathcal{O}(D_L^2 \times C) \tag{26}$$

Additionally, we must consider the computational cost of obtaining the predictions from the backbone architecture to generate pseudo-labels. For a single pixel, this complexity is:

$$\mathcal{O}(f(x^m_{h,w})) = \mathcal{O}(D_L) \tag{27}$$

The regularization term in the final pseudo-cross-entropy loss has a complexity of $\mathcal{O}(\|\beta\|^2) = \mathcal{O}(D_L)$, given that $\beta$ is of dimension $D_L \times 1$. The loss itself consists of linear operations, each with a computational complexity of $\mathcal{O}(1)$. Summing up these individual complexities, we get a computational complexity for processing a single image using the pseudo-cross-entropy loss:

$$\mathcal{O}(\mathcal{L}_{ce}(x^m)) = \mathcal{O}(D_L \times H \times W) \tag{28}$$

This leads us to comprehensive computational complexity for the cross-entropy portion of the loss function, for a single image, as:

$$\mathcal{O}((D_L^2 \times C + D_L + D_L) \times H \times W) = \mathcal{O}(D_L^2 \times C \times H \times W) \tag{29}$$

When using our combined loss function as described in Equation (2), the overall computational complexity for each minibatch, is as follows:

$$\mathcal{O}(\mathcal{L}_{combined}) \tag{30}$$
$$= \mathcal{O}(N \times (D_L \times C \times H \times W + C \times (HW)^2 + (HW)^2 + D_L^2 \times C \times H \times W)) \tag{31}$$
$$= \mathcal{O}(N \times C \times (HW)^2 + N \times D_L^2 \times C \times H \times W) \tag{32}$$

From now on, let $M \in \mathbb{N}$ be the number of parameters within the backbone architecture.

## C.2 TRAINING

The training algorithm is described in detail in Algorithm (1). During each minibatch iteration, the model updates not only the hidden-layer parameters $\{\mathbf{W}_1\}_{l=1}^{L-1}$, $\{\mathbf{b}_l\}_{l=1}^{L-1}$, but also the trainable output weights $\beta$. As calculated in Section C.1, the overall computational complexity required for each minibatch update using the complete loss function (2) is:

$$\mathcal{O}(M \times N \times C \times (HW)^2 + M \times N \times D_L^2 \times C \times H \times W) \tag{33}$$

Afterwards we perform spectral normalization using power iteration, as described in Liu et al. (2020). This introduces an additional complexity of:

$$\mathcal{O}(\sum_{l=1}^{L-1} D_l) \tag{34}$$

Lastly, we also update the precision matrix as indicated in Liu et al. (2020), leading to a complexity of:

$$\mathcal{O}(D_L^2) \tag{35}$$

Considering that the dimensions $\{D_l\}_{l=1}^{L-1}$ are fixed for a given architecture (e.g., for the STEGO architecture on the cityscapes data set $D_L = 100$), we get the aggregate complexity for each training minibatch:

$$\mathcal{O}(M \times N \times C \times (HW)^2 + M \times N \times D_L^2 \times C \times H \times W + \sum_{l=1}^{L-1} D_l) \tag{36}$$

## C.3 PREDICTION

The prediction algorithm is described in detail in Algorithm (2). During the prediction stage, for every pixel $x_{h,w}^m$, the model performs a forward pass computing the final hidden feature ($\phi$) and the posterior mean $g_k\left(f(x_{h,w}^m)\right) = \Phi^\top \beta_k$. Incorporating the computations involved in the backbone architecture, the complexity for this operation is:

$$\mathcal{O}(g_k\left(f(x_{h,w}^m)\right)) = \mathcal{O}(D_L \times C \times M) \tag{37}$$

Using the final hidden feature, we can compute the posterior variance matrices $\text{var}_k(\mathbf{x}) = \Phi^\top \hat{\Sigma}_k \Phi$, which incurs a computational complexity of:

$$\mathcal{O}(D_L^2) \tag{38}$$

Lastly, the predictive distribution $p_k$ is computed as $p_k = \exp(m_k) / \sum_k \exp(m_k)$, where $m_k \sim N\left(\hat{m}_k(\mathbf{x}), \hat{\sigma}_k^2(\mathbf{x})\right)$. We employ Monte Carlo Averaging to estimate the posterior mean of this distribution, following the methodology described in Liu et al. (2020). Given that this process is computationally friendly and only necessitates a single forward pass along with sampling a minimal set of samples, the total complexity for calculating $p_k$ is:

$$\mathcal{O}(p_k) = \mathcal{O}(D_L \times C \times M + D_L^2) \tag{39}$$

## C.4 QUALTITATIVE RESULTS

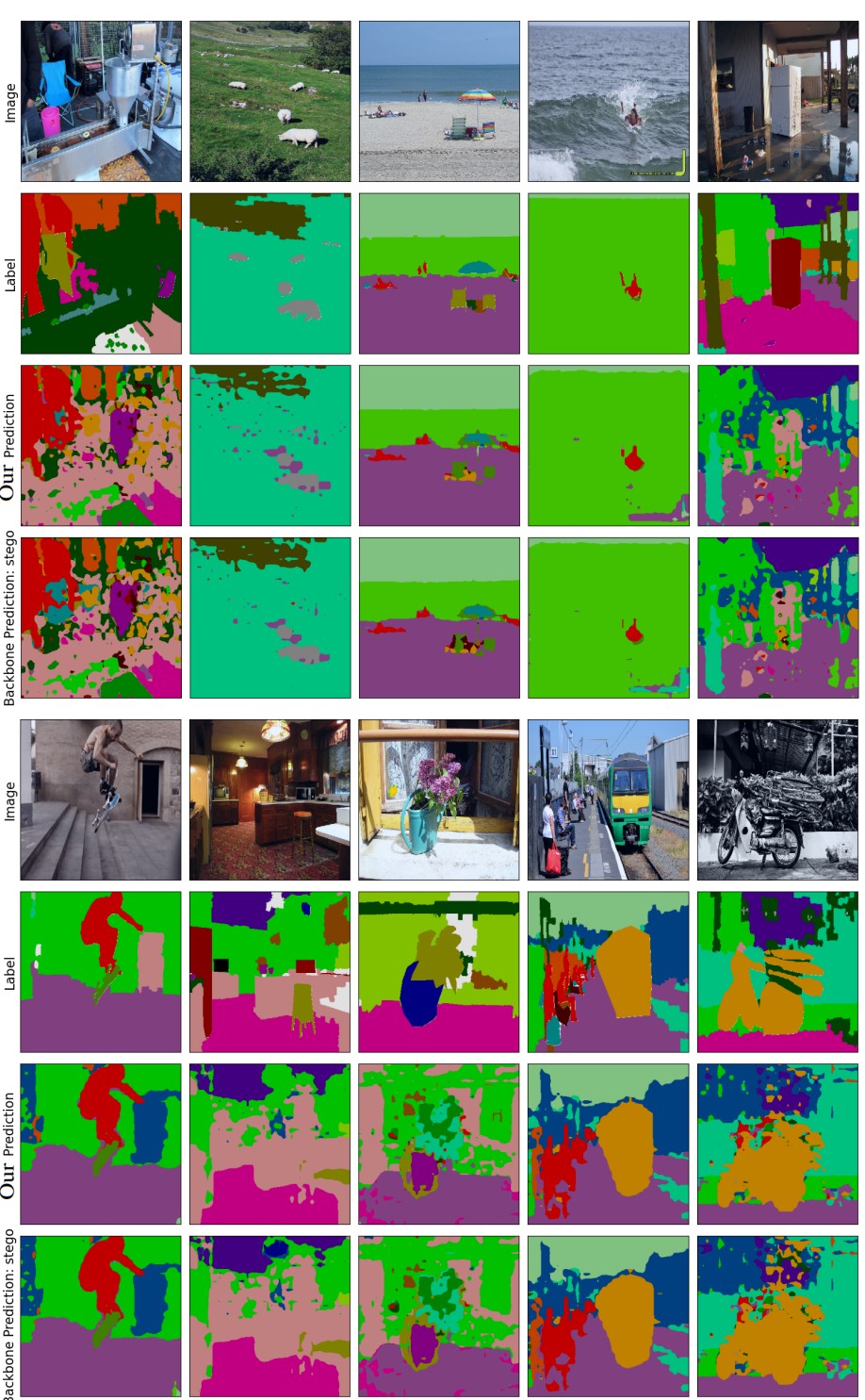

Figure 4: Comparison of ground truth labels ($2^{nd}$, $6^{th}$ row) and our predictions ($3^{rd}$, $7^{th}$ row), and backbone ($4^{th}$, last row) for images from the COCO-Stuff dataset.

