# OpenReview forum: "Robust Probabilistic Unsupervised Segmentation with Uncertainty Modeling"
_ICLR.cc/2025/Conference — Submitted to ICLR 2025_

### Official Review · Reviewer_tKnb · 2024-10-30

**Soundness:** 2
**Presentation:** 2
**Contribution:** 2
**Rating:** 3
**Confidence:** 4

**Summary:**

This paper presents a probabilistic framework for unsupervised semantic segmentation. It addresses limitations in traditional methods, such as class misalignment, spatial incoherence, and lack of uncertainty modeling, by incorporating uncertainty modeling and spatial smoothing. A new loss function is proposed to leverage feature information from pre-trained vision transformers to promote intra-class similarity, effectively managing complex textures and ambiguous regions. The approach demonstrates better accuracy and calibration on benchmarks like COCO, Potsdam, and Cityscapes.

**Strengths:**

- **Originality:** The concept of introducing uncertainty modeling into unsupervised semantic segmentation is intriguing.
- **Quality:** The development of the Spectral Normalized Gaussian Process (SNGP) is clearly presented and demonstrates a solid theoretical foundation.

**Weaknesses:**

- **Originality:** This work appears to be incremental, primarily integrating existing unsupervised semantic segmentation techniques with Spectral Normalized Gaussian Process (SNGP). Additionally, the proposed loss function closely resembles that of STEGO, while introduce more hyper-parameters.
- **Significance:** The performance improvements reported are modest. As shown in Tables 1, 2, and 3, the proposed method achieves similar or even lower results than existing approaches on unsupervised metrics (mIoU). In Table 4, combining this method with existing works results in minimal gains. This raises concerns that the performance increases are largely due to the additional tunable parameters rather than meaningful advancements.

**Questions:**

- In Figure 1, what is the diffience between patch embeddings $h$ and pixel embedding &h&?
- How many hyper-parameters are used in this work, and what methods were employed to tune them?
- If uncertainty modeling aims to improve segmentation in ambiguous regions, why are such simplistic examples shown in Figure 2? These visualizations do not effectively demonstrate its benefits, leading instead to questions about the model’s effectiveness in handling complex scenarios.
- The visualizations in the Appendix reveal significant noise in the predictions, calling into question the benefits of spatial smoothing techniques. SmooSeg [1], which uses a smoothness prior to enhance segment coherence and reduce noise, achieves marked improvements in prediction quality. Why was no comparison included with SmooSeg to better illustrate the advantages of the proposed smoothing approach?

_[1] Lan, M., Wang, X., Ke, Y., Xu, J., Feng, L. and Zhang, W., 2023. Smooseg: Smoothness prior for unsupervised semantic segmentation. Advances in Neural Information Processing Systems, 36, pp.11353-11373._

---

### Official Review · Reviewer_ZXK2 · 2024-11-03

**Soundness:** 2
**Presentation:** 2
**Contribution:** 2
**Rating:** 3
**Confidence:** 4

**Summary:**

The paper introduces a  probabilistic framework for unsupervised semantic segmentation that is uncertainty-aware. Each patch embedding is first pseudo-labelled. Then, a Spectral Normalized Gaussian Process is used to output an embedding for each pixel for clustering. The method outperforms various baselines across COCO, Potsdam, and Cityscape datasets.

**Strengths:**

Modelling uncertainty is an interesting direction for semantic segmentation.
The performance is quite competitive.

**Weaknesses:**

The presentation can be improved to enhance clarity. The introduction does not motivate well the method and does not state what proposed components are novel.  The term "SNGP" in line 97 is used without reference or explanation. The only crucial information in the introduction is that the paper proposes a novel loss to make the semantic segmentation framework uncertainty-aware without saying exactly what motivate the design of the loss and why it is novel. The contributions then are listed one after the other without proper motivations and contexts - which is very hard for readers to follow. I have no choice but to read thorough the method section to understand what the paper is about. However, it is also very technical without proper motivations for each introduced components. For example, why start from SNGP, why do we use pseudo-label, why do we use the segmentation head using a Gaussian Process (GP) with a Laplace approximation? I unfortunately find that in the method, we are just introducing different components of the framework one-by-one without any explanation. It lacks a logical flow that makes it hard to follow and understand what should I learn from the paper.

The way the paper is being organised also can be improved. In 3.4, the authors present the training loss with many unclear notations and then only explain them in section 4.2 (THEORETICAL JUSTIFICATION?). It is also unclear to me which loss term is novel or if the authors are referring to the combination of them being novel.

There is no analysis on the uncertainty of the pixel embedding and how they are helpful for challenging cases. The proposed method achieves quite competitive results on multiple datasets but it is unclear why. Some visualisation of successful cases in comparison to SOTA methods would help. Some ablation studies are very much needed to understand the effects of each proposed components.

**Questions:**

What are the novel components of the proposed method, and how do they contribute to the overall framework?

Why are specific techniques, such as SNGP and Gaussian Processes (GP) with Laplace approximation, chosen in the method, and what motivates their use?

What makes the proposed loss term novel, and is it the combination of terms or an individual term that introduces novelty?
Is there a clear theoretical justification for the proposed training loss and its components?


How does the proposed method quantify or analyze uncertainty in pixel embeddings, and how does this impact performance in challenging cases?

What is the individual impact of each proposed component on the performance of the method?

---

### Official Review · Reviewer_Eho2 · 2024-11-04

**Soundness:** 2
**Presentation:** 1
**Contribution:** 3
**Rating:** 5
**Confidence:** 3

**Summary:**

The paper presents a new method to perform unsupervised semantic segmentation, by proposing a probabilistic framework. The frameworks run on top of existing pseudo-labeling-based unsupervised semantic segmentation methods such as STEGO. The paper proposes a gaussian process-based post-processor that can be trained to remove noisy segmentation, especially on the edges, where the prior models would be more confused. The paper tries this on 3 datasets and shows the efficacy of these methods.

**Strengths:**

The paper presents a novel methodology to perform unsupervised semantic segmentation. Gaussian process-based post-processing has not been explored well in this area.

The quantitative results are solid and prove the efficacy of these methods.

**Weaknesses:**

The motivation needs to be framed better in the intro. It is unclear why an unsupervised segmentor needs to be uncertainty-aware. The introduction also lacks motivation for the method and how the proposed method can enable all these properties.

The method is hard to understand:
   * The paper presents a lot of jargon without going into details of how these techniques work as well as why they are needed. The paper for example does not introduce SNGP very well, making it unclear how it can be combined with the pseudo labels and image features.
   * The paper also introduces terms like random Fourier feature, and maximum correlation loss, without explaining what they mean. The maximum correlation loss is defined in a later section, after using the term. I would suggest a better way to organize the method section would be bottom-up and not top-down. i.e. first motivate and define each component and why it is needed and then finally combine them to produce equation 1 and 2.
   * A lot of places are missing citations as well for example SNGP, and RBG making it harder for the reader to even search for a background.
   * In a lot of places the equations are not rigorous and are incomplete. Several details are glossed over that should ideally not be glossed over. For example. in equation 1 what is $x, x^{knn}, x^{rand}$, where do the come from? See the questions section for more such clarifications. Similarly, the indexes, such as h,w,i,j, m, n, c are used without a definition.
   * The fine-grained segmentation (3.3) is not discussed before and it is unclear why/if it is needed. This component is not discussed in the ablation as well.
   * Overall, the paper needs to motivate what each component does and why is is needed.

The results are hard to interpret:
   * While the quantitative results show very clearly why the method works, the qualitative results are hard to interpret. The results shown in fig 2a and in supplementary figures, it is unclear if SNGP predictions are better than stego. If yes, if should be explained why they are better. An alternative could be to pick better examples, that show places where SNGP works better.
   * Similarly, in figure 2b. is is unclear why the variance images have these repeated grid patterns. If the method is truly measure uncertainty, we should not see such regular gridlike pattern, instead is should be smoother.
   * The ablation as well is a bit weak, it should ablate every component used, for example the three components of the loss function should be ablated, and the need of fine-grained segmentor should also be ablated.

**Questions:**

Minor suggestions
* In the intro the acronym SNGP is not defined before its first use.
* Related work on SNGP can be mentioned in more papers, such as instances of SNGP being used for other tasks in CV and ML that require uncertainty estimation.
* In equation 2, $\mathcal{L}_{total}$ in the LHS should take in arguments to be well-defined.
* What is w in equation 3?
* In eq 5 is the n on the LHS supposed to be N?

---

### Official Review · Reviewer_xEmq · 2024-11-11

**Soundness:** 2
**Presentation:** 2
**Contribution:** 2
**Rating:** 3
**Confidence:** 4

**Summary:**

This work introduces a probabilistic framework for unsupervised semantic segmentation that enhances robustness and accuracy by modeling uncertainty and employing spatial smoothing techniques. A new loss function that leverages feature information from pre-trained ViTs to learn similarities within pixels is proposed. The proposed method outperforms various baselines across multiple unsupervised semantic segmentation benchmarks, including COCO, Potsdam, and Cityscapes, showing superior accuracy and calibration.

**Strengths:**

1. The proposed uncertainty modeling is a significant strength to model prediction uncertainty, especially in ambiguous regions, leading to more reliable segmentation results.

2. The introduction of a new loss function that encourages learning similarities within pixels is innovative and contributes to the improved performance of the model.

3. The paper provides theoretical analyses forthe proposed algorithm, which is crucial for understanding the robustness and effectiveness of the approach.

4. SOTA performance and seamlessly integrate with existing methods.

**Weaknesses:**

1. The use of higher-dimensional probabilistic embeddings, while improving performance, may increase computational costs and the risk of overfitting, which could be a limitation. This point is mentioned at the end of the article, but it is better to give some experimental results and solutions.

2. The complexity of the model, due to the probabilistic framework and the additional loss function, might make it more challenging to train and optimize compared to simpler models.

3. The paper does not extensively discuss the framework's generalization ability to datasets outside of the tested benchmarks, which is crucial for assessing its real-world applicability.

4. Minor comments:
- Incomplete reference such as line 766, line 810, line 818 Equation ??.
- Some of the content in the appendix can be put into the body to ensure that the body is a full 10 pages.

**Questions:**

1. How does the framework scale with increasing image resolution and dataset size, and what are the corresponding resource requirements?

2. How sensitive is the model's performance to changes in hyperparameters, particularly those related to the loss function?

3. How does the framework perform in real-world applications, especially in scenarios like medical imaging where robust segmentation is critical?

4. What is the test time for the model compared to other methods, and how does it scale with the size of the dataset?

---

### Meta-Review · Area_Chair_Wnhz · 2024-12-20

**Metareview:**

All four reviewers held a unanimous opinion to reject this paper, and the authors did not provide a rebuttal.
While it is intriguing to investigate uncertainty for unsupervised semantic segmentation, the motivation is not entirely clear. This paper is difficult to read as several reviewers pointed out. The pipeline here relies on multiple components, many of which are not discussed in sufficient detail nor related papers were properly referred to. It looks like the authors gave theoretical justification but did not clearly describe the method in the first place, which make the reviewers not appreciate the organization of the paper.
Reviewers are also concerned about the lack of contribution, as the method here chains several components including SNGP and STEGO.

The authors are encouraged to improve the presentation and writing of this paper and clearly articulate their motivation and contributions w.r.t. prior work.

**Additional Comments On Reviewer Discussion:**

No rebuttal was provided by the authors and no discussion among reviewers.

---

### Decision · Program_Chairs · 2025-01-22

Reject